# Discovery of New VEGFR-2 Inhibitors: Design, Synthesis, Anti-Proliferative Evaluation, Docking, and MD Simulation Studies

**DOI:** 10.3390/molecules27196203

**Published:** 2022-09-21

**Authors:** Eslam B. Elkaeed, Reda G. Yousef, Mohamed M. Khalifa, Albaraa Ibrahim, Ahmed B. M. Mehany, Ibraheem M. M. Gobaara, Bshra A. Alsfouk, Wagdy M. Eldehna, Ahmed M. Metwaly, Ibrahim H. Eissa, Mohamed Ayman El-Zahabi

**Affiliations:** 1Department of Pharmaceutical Sciences, College of Pharmacy, AlMaarefa University, Riyadh 13713, Saudi Arabia; 2Pharmaceutical Medicinal Chemistry & Drug Design Department, Faculty of Pharmacy (Boys), Al-Azhar University, Cairo 11884, Egypt; 3Zoology Department, Faculty of Science (Boys), Al-Azhar University, Cairo 11884, Egypt; 4Department of Pharmaceutical Sciences, College of Pharmacy, Princess Nourah bint Abdulrahman University, P.O. Box 84428, Riyadh 11671, Saudi Arabia; 5Department of Pharmaceutical Chemistry, Faculty of Pharmacy, Kafrelsheikh University, Kafrelsheikh 33516, Egypt; 6School of Biotechnology, Badr University in Cairo, Badr City 11829, Egypt; 7Pharmacognosy and Medicinal Plants Department, Faculty of Pharmacy (Boys), Al-Azhar University, Cairo 11884, Egypt; 8Biopharmaceutical Products Research Department, Genetic Engineering and Biotechnology Research Institute, City of Scientific Research and Technological Applications (SRTA-City), Alexandria 21934, Egypt

**Keywords:** apoptosis, anti-proliferative, immunomodulation, MD simulations, nicotinamide, VEGFR-2 inhibitors

## Abstract

Four new nicotinamide-based derivatives were designed as antiangiogenic VEGFR-2 inhibitors. The congeners were synthesized possessing the pharmacophoric essential features to bind correctly with the VEGFR-2 active pocket. All members were evaluated for their cytotoxic and VEGFR-2 inhibitory potentialities. Compound **6** was the most potent showingIC_50_ values of 9.3 ± 0.02 and 7.8 ± 0.025 µM against HCT-116 and HepG-2 cells, respectively, and IC_50_ of 60.83 nM regarding VEGFR-2 enzyme inhibition. Compound **6** arrested the growth of HCT-116 cells at the pre-G1 and G2-M phases. Further, it induced both early and late apoptosis. Additionally, compound **6** caused a significant decrease in TNF-*α* and IL6 by 66.42% and 57.34%, respectively. The considered compounds had similar docking performances to that of sorafenib against the VEGFR-2 (PDB ID: 2OH4). The correct binding of compound **6** with VEGFR-2 was validated using MD simulations, and MM-GPSA calculations.

## 1. Introduction

Based on the WHO estimates, cancer is the 2^nd^ highest cause of death in about 112 countries, while it is considered the 3^rd^ to the 4^th^ highest in about 23 countries. By 2020, the total number of new cancer cases was estimated to be almost 9 million. This number is predicted to increase to 28 million cases by 2040 [1]. In response, many efforts were made to generate efficient and low-toxic anticancer agents [2].

Upregulated angiogenesis is a key feature of the growth of several cancer types [3]. Angiogenesis, the process by which new blood capillaries are developed from the existing vasculature, is stimulated through the activation of various chemical signals [4]. Tyrosine kinases (TKs) are main regulators of tumor angiogenesis. The receptors of TKs, especially VEGFR-2, are over-expressed in different cancer cells [5]. In response to its activation, VEGFR-2 promotes a series of successive signals that improve cell survival, growth, and proliferation [5]. The over-activity of VEGFR-2 receptors was reported in the cancer cells versus the normal cells. This fact enabled researchers to target them therapeutically to produce safe and selective drugs that tackle angiogenesis in tumor cells with no activity on normal cells. The strategy to hinder the VEGF pathway is carried out by blocking the VEGFR-2 receptors activation using VEGFR-2 inhibitors. 

Several FDA-approved VEGFR-2 inhibiting drugs were introduced to the market in the past few decades. These drugs have been recognized to control different cancer types *via* inhibiting tumor angiogenesis. On the dark side, many side effects have happened parallel to the treatment programs of VEGFR-2 inhibitors. These drawbacks opened the door of searching for new small molecules with potential effects and fewer side effects [6]. The in silico/computational chemistry methods were conducted as successful tools in drug design and discovery [7,8], and ADMET [9] analysis of new drugs. 

Different pyridine-based compounds are well defined recently as VEGFR-2 inhibitors [10,11,12]. Sorafenib, **1**, is one of the most important pyridine-based anti-angiogenic compounds and widely used in the management of hepatocellular carcinoma and breast cancer [13]. Sorafenib acts as an allosteric inhibitor at the TK active site. The sorafenib’s distal hydrophobic moiety (hydrophobic tail) is primarily directed toward the allosteric lipophilic pocket, converting the enzyme to an inactive form. Consequently, the urea (pharmacophore) moiety of sorafenib occupies the DFG motif region forming hydrogen bond (H-bond) interactions with Asp1044 and Glu883 [14]. Furthermore, the receptor’s adenine region (hinge region) is filled with the pyridine motif of sorafenib (forming two H-bond interactions with Cys917 [15], and the gatekeeper region of the receptor is occupied by the central phenyl ring (linker) of sorafenib (Figure 1). Despite its high activity and selectivity, sorafenib still possesses some pharmacokinetic problems represented in either its bad water solubility or decreased oral bioavailability. The later problems potentiated the need for the discovery of new VEGFR-2 anti-angiogenic drugs.

Over the recent years, our lab members have developed a project aiming to overcome the later challenges by increasing or, at least, conserving the activity with the enhancement of the pharmacokinetic properties. We designed, synthesized and examined the anti-VEGFR-2 activities of various compounds containing benzoxazole [16], pyridine [17] quinazoline [18], quinazoline [19], thiourea-azetidine [20,21] and quinoxaline-2 (1*H*)-one [22], in addition to thieno[2,3-*d*]pyrimidine scaffolds [23]. 

Continuously, we tried to discover more derivatives in the hope of obtaining new VEGFR-2 inhibitors. The hypothesis of the present work is to build four new nicotinamide-based congeners whose structures are consists of four main parts. The 1st part is the proximal nicotinamide moiety that binds to the hinge region (adenine binding), and the 2^nd^ part is a phenyl group that acts as a linker. The hydrogen-bonding moiety (the 3^rd^ part) of our compounds was decided to be either a hydrazino carbonyl moiety or a carbamoyl hydrazineylidene moiety. Different distal hydrophobic tails (the 4^th^ part) were chosen to target the allosteric site (Figure 2). 

The target compounds were designed to have better aqueous solubility than sorafenib and consequently good oral bioavailability. This objective was achieved through two strategies. The first is the replacement of the urea moiety of sorafenib with either hydrazino carbonyl or carbamoyl hydrazineylidene moieties. These moieties have higher hydrophilicity levels that should increase the chance of water solubility. The second strategy is the incorporation of hydrophilic groups at the hydrophobic tails of the target compounds. In detail, a hydroxyl group was incorporated in the hydrophobic tail of compound **7**, an NH group was incorporated in the hydrophobic tail of compound **8**, and an NH_2_ group was incorporated in compound **10**. The degrees of water solubility of the target compounds as well as sorafenib were tested in silico.

## 2. Results and Discussion

### 2.1. Chemistry

The targeted members were synthesized as outlined in Figure 1 and Figure 2. At first, nicotinic acid **2** was chlorinated using thionyl chloride yielded nicotinoyl chloride **3** [24]. The latter compound **3** was then condensed with ethyl-4-aminobenzoate yielding ethyl 4-(nicotinamido)benzoate **4,** which was then refluxed with hydrazine hydrate to afford the corresponding acid hydrazide **5**. Following its crystallization, compound **5** was reacted with different aldehydes; namely, 4-chlorobenzaldehyde, 2-hydroxybenzaldehyde, and 1*H*-indole-3-carbaldehyde, to give the corresponding benzylidene derivatives **6**, **7**, and **8**, respectively (Figure 1). 

Spectral data of the latter compounds confirmed their structures. However, ^1^H NMR of member **7**, a representative example of this series, showed the proton of the OH at *δ* 11.37 ppm as a singlet signal.

Furthermore, compound **4** was reacted with semicarbazide **9** to produce the corresponding member **10**. Compound **10** was confirmed with different spectral data, as its ^1^H NMR exhibited the NH_2_ protons at *δ* of 6.5 ppm as a singlet signal. Additionally, the IR spectrum showed the absorption bands of the NH and NH_2_ groups at 3406 and 3326 cm^−1^.(Figure 2).

### 2.2. Biological Testing

#### 2.2.1. In Vitro Anti-proliferative Activities against HepG-2 and HCT-116

The four synthesized nicotinamide derivatives as well as sorafenib were in vitro screened for their cytotoxic potentialities against hepatocellular (HepG-2) and colorectal carcinoma (HCT-116). The latter cells were selected owing to their VEGFR-2 overabundance. It was obvious from the achieved results that the obtained congeners possess varying selectivity degrees against the examined cell lines in comparison to the reference drug. However, the chlorobenzylidene congener, **6**, showed the best cytotoxic activity among the synthesized congeners, with IC_50_ values of 7.80 ± 0.025 and 9.3 ± 0.02 µM against hepatocellular and colorectal cells, respectively. Higher IC_50_ values were noticed upon replacing the chlorobenzylidene moiety with the hydroxybenzylidene in compound **7**, as it showed IC_50_ values of 10.20 ± 0.035 for HepG-2 and 15.9 ± 0.041 µM for HCT-116. On the other side, compounds **8** and **10** exhibited moderate activities against both cells with IC_50_ values ranging from 16.03 ± 0.051 to 24.2 ± 0.06 µM (Table 1).

#### 2.2.2. In Vitro VEGFR-2 Enzyme Assay

The ability of the four synthesized congeners to inhibit the VEGFR-2 enzyme in HCT-116 cells was also evaluated. Results were then compared to sorafenib. A clear conclusion was obtained from the outputted results, as members **6** and **10** gave the best inhibitory effects with IC_50_ values of 60.83 and 63.61 nM, respectively, the values that were almost equal to that of sorafenib (IC_50_ = 53.65 nM). In contrast, a moderate inhibitory effect was observed regarding compound **7** with an IC_50_ value of 129.30 nM (Table 1).

#### 2.2.3. Effect on Cell Cycle Phases

The cytotoxicity results and the VEGFR-2 inhibitory assessment of the new compounds encouraged us to investigate the effect of member **6** on the cell cycle progression of HCT-116 cells as demonstrated in Table 2 and Figure 3**.**

It was clear from the obtained results that member **6** arrested cell cycle progression at the phase G2-M, as it caused a significant elevation of the cell levels by 35.84% versus 12.91% accumulation of the control cells. Compound **6**, moreover, caused a considerable increase in the apoptotic cells at the pre-G1 phase (16.93%) comparing the control cells (3.05%). The former results in addition to the decrease in the S phase percentage of the treated cells to 27.45% confirmed the ability of **6** to arrest HCT-116 cells progression.

#### 2.2.4. Detection of Apoptosis

Both extrinsic and intrinsic apoptosis induced by compound **6** in HCT-116 cells was assayed through Annexin-V/propidium iodide staining assay. Marked induction of both early and late apoptosis was observed upon treating HCT-116 cells with **6**. As the percentages of early and late apoptosis increased from 0.70% and 1.73% in the control cells to 5.78% and 9.75% in compound **6** treated cells. Results of apoptosis induction were displayed in Table 3 and Figure 4. 

#### 2.2.5. In Vitro Immunomodulatory Assay

The immunomodulatory potentialities of congeners **6** and **7** on HCT-116 cells was also assayed. Two well-defined immunity markers, human tumor necrosis factor alpha (TNF-*α*), and interleukin 6 (IL6), were measured following the treatment with each compound. In addition, dexamethasone was co-assayed as a reference drug. A considerable reduction in both markers was noticed regarding the two compounds (Table 4). In detail, compound **6** caused a decrease in TNF-*α* and IL6 by 66.42% and 57.34%, respectively. Additionally, compound **7** reduced TNF-**α** and IL6 to 60.54% and 52.75%, respectively.

### 2.3. In Silico Studies 

#### 2.3.1. Molecular Docking 

In silico docking studies were then performed to clarify the key points of interaction between the synthesized members and the VEGFR-2 ATP pocket. The VEGFR-2 structure (PDB: 2OH4) and its native ligand, a benzimidazole-urea inhibitor, were selected for the intended study. Thus, the protein crystal structure was downloaded and prepared using MOE software. Following its preparation, a validation step was performed *via* re-docking of the native ligand onto the active pocket, the step that confirmed the suitability of the planned docking protocol by reproducing an identical binding pattern to that of the downloaded ligand Figure 5. However, a detailed binding pattern of the benzimidazole-urea inhibitor was illustrated in Figure 6.

Results of docking studies indicated that congener **6** is bound tightly to the active pocket *via* the engagement in three H-bonds. One H-bond was formed between the nicotinamide moiety and Cys917 residue in the hinge region, while the other two H-bonds were formed between the pharmacophoric hydrazinyl moiety and the essential amino acids Asp1044 and Glu883 in the DFG motif (Figure 7). Moreover, different hydrophobic interactions (HIs) supported the proposed binding, as **6** formed three HI interactions with Phe916 and Ile886. These findings indicated that compound **6** has a binding pattern similar to that of the benzimidazole-urea inhibitor. Additionally, the binding free energy of compound **6** is −28.52 kcal/mol, which was very close to that of the benzimidazole-urea inhibitor (−28.02 kcal/mol) (Table 5).

On the other hand, a superior binding pattern was observed regarding compound **7**. As it bonded to the ATP active pocket through five H-bonding interactions. Like **6**, the nicotinamide moiety of **7** bound to Cys917 via an H-bond and the hydrazinyl moiety formed two H-bonds with Asp1044 and Glu883 residues. Moreover, the pharmacophoric group formed two H-bonds with Lys886 through its carbonyl group. Several HI interactions also strengthened the binding mode. The central phenyl moiety formed three different HI interactions with Phe1045, Ile886, and Val897 amino acids, while the terminal phenyl moiety bonded by a π–π interaction with Phe916 (Figure 8).

#### 2.3.2. MD Simulations

Molecular docking studies are a sort of structure-based studies that have an essential drawback of describing the protein’s interaction with ligand as a rigid unit. Therefore, docking experiments do not examine or identify the conformational changes in protein structure after the binding of the active compound (ligand) [25]. Meanwhile, MD simulations experiments can be used to investigate accurately and evaluate precisely, at an atomic resolution, the behavior and changes in the protein’s structure that occurred after binding [26].

To analyze the stability after binding, in addition to the dynamic changes of the compound **6**-VEGFR-2 complex, the occurred conformational changes have been investigated for compound **6**, VEGFR-2, and compound **6**-VEGFR-2 complex from the perspective of RMSD (Figure 9A). The compound **6**-VEGFR-2 complex had low RMSD values featuring no obvious fluctuations following the binding over the MD study. Likewise, we examined the flexibility of the VEGFR-2 in respect of RMSF calculation to predict the changes in the regional flexibility in the MD simulations experiment. The experiment showed that compound **6**’s binding of the VEGFR-2 makes the second flexible slightly in 1050–1070 residue areas (Figure 9B). Regarding the radius of gyration (R_g_) parameter, it refers to the change in protein volume that determines the stability of an enzyme. R_g_ is the RMSD of a weighted unit of mass of atoms away from its mass center. The R_g_ parameter identifies the 3D changes in a protein in addition to the compactness. The level of fluctuation through the simulations experiment is inversely proportional to the compactness and stability of the complex [27,28]. The computed R_g_ of the compound **6**-VEGFR-2 complex (Figure 9C) was found to be stable and less than the initial time confirming the compactness and stability. Solvent accessible surface area, SASA, was further used to compute the interaction of the compound **6**-VEGFR-2 complex with the encompassing solvents during the simulation (100 ns) time, and it revealed the changes in conformation during the simulation. Importantly, the SASA values for the compound **6**-VEGFR-2 complex (Figure 9D) were considerably lower than the beginning values, indicating the reduction of surface area and correspondingly the stability and integrity of the compound **6**-VEGFR-2 complex. Since hydrogen bonding between a complex is essential for stabilizing it, MD simulation experiments were conducted to investigate hydrogen bonding through the compound **6**-VEGFR-2 complex and exhibited that the highest conformations of HBs in the formed complex was up to four HBs.

Figure 10 illustrates the conformational analysis of compound **6** inside the active site of the VEGFR-2 enzyme during the first (Figure 10A) and 100^th^ (Figure 10B) nanoseconds (ns) of the MD production run, showing that conformational changes occurred. More interestingly, this study confirms the binding stability and the integrity of the compound **6**-VEGFR-2 complex, as it shows that compound **6** remained bound tightly to the active site over the 100 ns.

#### 2.3.3. MM-PBSA Studies

Through the molecular mechanics energies combined and the Poisson–Boltzmann surface area continuum solvation, MM-PBSA, the binding of a receptor and compound can be determined via the computation of the binding exact free energy of the compound–protein complex over the simulation period. The MM-PBSA relies on molecular dynamics (MD) and thermodynamic cycle methods. In order to determine the binding energy correctly, two types of energies have to be evaluated. These energies are gas-phase interaction energy (electrostatic interactions and van der Waals ) and solvation energy (polar and non-polar components) [29]. 

The binding free energy of the compound **6**-VEGFR-2 complex was measured from MD trajectories employing the MM/PBSA method at the last 20 ns of the production run at an interval of 100 ps. Compound **6** demonstrated an excellent binding free energy of −125 KJ/mol with VEGFR-2. Significantly, the binding energy was stable (Figure 11A) throughout the 20 ns of the study, indicating the compound **6**-VEGFR-2 complex was accurately bound.

Next, the binding free energy was disassembled to determine the various components of the binding energy obtained besides the exact contribution of every amino acid in the binding process. Seven amino acid residues (LEU-838, GLU-883, VAL-914, PHE-916, CYS-917, CYS-1043, and PHE-1045) in VEGFR-2 (Figure 11B) contributed more than −4 KJ/mol binding energy and were considered vital (essential) residues in the interaction.

#### 2.3.4. In Silico ADMET Analysis

The pharmacokinetics profile of the considered compounds was evaluated computationally via ADMET studies using Discovery Studio 4.0 software. The FDA-approved sorafenib was employed as a reference. The calculated results of ADMET studies were summarized in Table 6.

All compounds expressed predicted a low ability in BBB penetration power (BBB-P), indicating the reduced CNS side effect. Compound **10** showed an optimal solubility level (Sol-L), and compound **7** demonstrated a good solubility level. Compounds **6** and **8** were predicted to have poor solubility levels, but still better than sorafenib. Regarding intestinal absorption (Int-A), all the synthesized members were estimated to have good levels. For metabolic investigation, all the synthesized members were predicted to be non-inhibitors of the cytochrome-P450, (CYP2D6-Inh). Furthermore, compounds **6** and **7** were anticipated to bind by more than 90% to the plasma protein (PP-Bind). In contrast, compounds **8** and **10** were expected to bind by less than 90% (Figure 12).

#### 2.3.5. Toxicity Studies 

Toxicity profiles of the synthesized compounds were calculated from Discovery studio software version 4.0 [30,31]. This profile includes six models: carcinogenic potency TD_50_ (R TD_50_) [32], rat maximum tolerated dose (R MTD) [33,34], rat oral LD_50_ [35], rat chronic lowest observed adverse effect level (R LOAEL [36,37], skin irritancy, and ocular irritancy.

For the R TD_50_, compounds **7**, **8**, and **10** were predicted to have TD_50_ values of 276.321, 14.966, and 54.777 g/kg, respectively. These values are more than that of sorafenib (14.245 g/kg). In addition, all the tested compounds were predicted to have R MTD ranging from 0.144 to 0.414 g/kg, more than that of sorafenib (0.089 g/kg). Except for compound **10**, all the tested compounds, **6**, **7**, and **8**, showed higher values of LD_50_ (R LD_50_) than that of sorafenib (0.823 g/kg). In addition, all the tested members were predicted to have R LOAEL values (ranging from 0.096 to 0.481 g/kg) higher than that of sorafenib (0.005 g/kg). Furthermore, the tested compounds were anticipated to have a mild irritancy effect against the eye with diminished irritancy against the skin (Table 7).

## 3. Materials and Methods

### 3.1. Chemistry

#### 3.1.1. General

In-depth discussions of reagents, chemicals, and apparatuses have been held in the Appendix A. Compounds **3**, **4**, and **5** were reported [8,38] before. The synthesized compounds were analyzed using IR and NMR spectroscopy. The IR was carried out using KBr disc method at cm^−1^. The ^1^H NMR was carried out at 400 MHz using DMSO-*d*6 as solvent. The ^1^H NMR was carried out at 100 MHz using DMSO-*d*6 as solvent.

#### 3.1.2. Synthesis of Congeners **6**, **7**, and **8**

Compound **5** (0.256 g, 0.001 mol) was mixed with three aromatic aldehydes (0.001 mol each) namely, 4-chlorobenzaldehyde, 2-hydroxybenzaldehyde, and 1*H*-indole-3-carbaldehyde. The mixtures were boiled in of absolute ethanol (30 mL) with g CH_3_COOH (a few drops) for 2 h. Then, the cooled, filtered, dried, and obtained precipitates were crystallized from ethanol to afford compounds **6**, **7**, and **8**, respectively.

##### N-{4-[2-(4-Chlorobenzylidene)hydrazine-1-carbonyl]phenyl}nicotinamide **6**

Yield: 83%; Melting point: 266–268 °C; IR: 3327, 3260, 1654; ^1^H NMR: 7.52 (d, *J* = 8.0 Hz, 2H), 7.82, (dd, *J* = 8.0, 8.0 Hz, 1H), 7.75 (d, *J* = 8.0 Hz, 2H), 7.93 (m, 4H), 8.31 (dt, *J* = 8.0, 8.0, 8.0 Hz, 1H), 8.46 (s, 1H), 8.78 (dd, *J* = 8.8, 8.8 Hz, 1H), 9.14 (s, 1H), 10.71 (s, 1H), 11.88 (s, 1H); ^13^C NMR: 120.06, 124.02, 124.02, 128.76, 129.01, 129.15, 129.42, 129.42, 130.81, 133.83, 134.91, 136.05, 136.05, 142.35, 146.08, 149.24, 149.24, 152.24, 163.01, 164.90; Anal. Calcd. For C_20_H_15_ClN_4_O_2_ (378.82): C, 63.41; H, 3.99; N, 14.79. Found: C, 63.69; H, 4.15; N, 15.08%. 

##### N-{4-[2-(2-Hydroxybenzylidene)hydrazine-1-carbonyl]phenyl}nicotinamide **7**

Yield: 79%; Melting point: 238–240 °C; IR υ_max_/cm^−1^: 3335, 3259, 1658; ^1^H NMR: 6.94 (m, 2H), 7.31 (t, *J* = 7.6 Hz, 1H), 7.55 (d, *J* = 7.6 Hz, 1H), 7.60 (dd, *J* = 8.0, 8.0 Hz, 1H), 7.96 (d, *J* = 8.8 Hz, 2H), 8.00 (d, *J* = 8.8 Hz, 2H), 8.35 (d, *J* = 8.0 Hz, 1H), 8.66 (s, 1H), 8.79 (s, 1H), 9.15 (s, 1H), 10.75 (s, 1H), 11.37 (s, 1H), 12.11 (s, 1H); ^13^C NMR: 116.90, 119.18, 119.81, 120.11, 120.11, 124.02, 128.24, 129.04, 129.04, 130.04, 130.81, 131.80, 136.07, 142.70, 148.54, 149.26, 152.81, 157.95, 162.70, 164.93; Anal. Calcd. For C_20_H_16_N_4_O_3_ (360.37): C, 66.66; H, 4.48; N, 15.55. Found: C, 66.90; H, 4.67; N, 15.69%. 

##### N-{4-[2-((1H-Indol-3-yl)methylene)hydrazine-1-carbonyl]phenyl}nicotinamide **8**

Yield: 74%; Melting point: 247–249 °C; IR: 3263, 3201, 1657; ^1^H NMR: 7.15 (t, *J* = 7.6 Hz, 1H), 7.24 (t, *J* = 8.0 Hz, 1H), 7.46 (d, *J* = 8.0 Hz, 1H), 7.59 (dd, *J* = 7.6, 7.6 Hz, 1H), 7.84 (s, 1H), 7.94 (d, *J* = 9.2 Hz, 2H), 7.99 (d, *J* = 9.2 Hz, 2H), 8.34 (m, 2H), 8.66 (s, 1H), 8.93 (dd, *J* = 6.8, 6.8 Hz, 1H), 9.16 (s, 1H), 10.74 (s, 1H), 11.54 (s, 1H), 11.62 (s, 1H); ^13^C NMR: 112.26, 112.30, 120.10, 120.86, 122.51, 123.11, 123.95, 124.03, 124.86, 128.78, 129.59, 130.70, 130.87, 136.07, 137.52, 142.10, 145.24, 149.25, 152.78, 162.46, 164.88, 185.48; Anal. Calcd. For C_22_H_17_N_5_O_2_ (383.41): C, 68.92; H, 4.47; N, 18.27. Found: C, 68.74; H, 4.58; N, 18.48%.

##### N-{4-[1-(2-Carbamoylhydrazineylidene)ethyl]phenyl}nicotinamide **10**

Compound 4 (0.24 g, 0.001 mol) and semicarbazide **9** (0.07 g, 0.001 mol) were refluxed in ethanol (30 mL) in the presence of a few drops of CH_3_COOHfor 6 h. After finishing point, the mixture was cooled and filtered. The formed precipitate was crystallized from ethanol to produce compound **10**. 

Yield: 72%; Melting point: 246–248 °C; IR: 3406, 3326, 1702, 1656; ^1^H NMR: 2.18 (s, 3H), 6.50 (s, 2H), 7.57 (m, 1H), 7.79 (d, *J* = 8.8 Hz, 2H), 7.86 (d, *J* = 8.8 Hz, 2H), 8.29 (dt, *J* = 8.0 Hz, 1H), 8.76 (2d, *J* = 4.8 Hz, 1H), 9.11 (dd, 1H), 9.29 (s, 1H), 10.51 (s, 1H); ^13^C NMR: 13.55, 120.20, 120.20, 123.97, 126.86, 126.86, 130.99, 134.22, 135.95, 139.57, 144.06, 149.84, 153.64, 157.81, 164.53; Anal. Calcd. For C_15_H_15_N_5_O_2_ (297.32): C, 60.60; H, 5.09; N, 23.56. Found: C, 60.87; H, 5.31; N, 23.72%.

### 3.2. Biological Testing

#### 3.2.1. In Vitro Anti-proliferative Activities against HepG-2 and HCT-116

Was conducted the experiment by the MTT procedure [39,40]. In-depth discussions have been held in the Appendix A. 

#### 3.2.2. In Vitro VEGFR-2 Enzyme Inhibition Assay 

Was conducted the experiment by Human VEGFR-2 ELISA kit [41]. In-depth discussions have been held in the Appendix A. 

#### 3.2.3. Flow Cytometry Analysis for Cell Cycle 

Were conducted propidium iodide (PI) staining and flow cytometry analysis. [42,43]. In-depth discussions have been held in the Appendix A.

#### 3.2.4. Flow Cytometry Analysis for Apoptosis

In-depth discussions have been held in the Appendix A [44,45].

### 3.3. In Silico Studies 

#### 3.3.1. Docking Studies 

Docking studies against VEGFR-2 [PDB: 2OH4] resolution: 2.03 Å were conducted by MOE2014 software (Chemical Computing Group Inc., Quebec, Canada). In-depth discussions have been held in the Appendix A. 

#### 3.3.2. ADMET Studies

The studies were conducted by Discovery studio 4.0 [46]. In-depth discussions have been held in the Appendix A.

#### 3.3.3. Toxicity Studies 

The studies were conducted by Discovery studio 4.0. In-depth discussions have been held in the Appendix A [47].

#### 3.3.4. Molecular Dynamics Simulation & MM/PBSA

MD simulation experiments and MM/PBSA were conducted by GROMACS [48,49,50,51]. In-depth discussions have been held in the Appendix A.

## 4. Conclusions

To conclude, a new series of nicotinamide-based derivatives was designed as antiangiogenic VEGFR-2 inhibitors. Four congeners were synthesized. The four members possessed the key features to bind with the VEGFR-2 active pocket. Biologically, all members were evaluated for their cytotoxic effects regarding HCT-116 and HepG-2 cells. Additionally, in vitro VEGFR-2 inhibitory effects were assayed for the synthesized series. The results of the former tests revealed that member **6** was the most potent among the tested compounds, with IC_50_ values of 9.3 ± 0.02 and 7.8 ± 0.025 µM against HCT-116 and HepG-2 cells, respectively, and IC_50_ of 60.83 µM regarding VEGFR-2 enzyme. Compound **6** strongly arrested the cell at pre-G1 and G2-M phases and induced apoptosis at early and late stages. Additionally, it decreased TNF-α and IL6 by 66.42% and 57.34%, respectively, pointing to a potent immunomodulatory effect. Computationally, the synthesized compounds showed similar docking demeanors to sorafenib against VEGFR-2 (PDB ID: 2OH4). The MD simulations experiments validated the correctness binding of compound **6** over 100 ns. Additionally, the MM-PBSA analysis verified the optimum binding with excellent energy. Finally, ADMET profiling indicated the general likeness and safety of the considered compounds.

## Data Availability

Not applicable.

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
