# Peer review of "Discovery of New VEGFR-2 Inhibitors: Design, Synthesis, Anti-Proliferative Evaluation, Docking, and MD Simulation Studies"

_molecules, 2022, doi:10.3390/molecules27196203_

Round 1
Reviewer 1 Report
This manuscript described, design, synthesis, antiproliferative evaluation, docking, and MD simulation studies of 4 new VEGFR-2 inhibitors. Based on the Biological testing in the manuscript, these 4 new pounds did not show promising result, compared to sorafenib data. Based on the manuscript, I have few questions listed here.
1. Compound 6, compound 10, C13 NMRs should be adjusted the phase line, to make the peaks to be positive position.
2. Compound 6 C13 NMR should show 0ppm to 200ppm, H NMR should show 0ppm to 14ppm.
3. Compound 7 NMR in supporting information, 2.5ppm was DMSO solvent peak, 3.3ppm was the H2O residual, they should not do the integration.
4. Compound 8 NMR in supporting information, integration should be integer
5. Page 2, line 76 to line 79, authors described “sorafenib still 76 possesses some pharmacokinetic problems represented in either its bad water solubility or decreased oral bioavailability. The later problems potentiated the need for the discovery of new VEGFR-2 anti-angiogenic drugs.” But in the manuscript, the design part, did not show the logic, to solve the solubility and oral bioavailability problem. Manuscript better describe it, how authors designed the 4 compounds. Especially, the title was “Design and synthesis”.
6. In vitro VEGFR-2 enzyme assay inhibition showed the new 4 compounds were not better than sorafenib. The compounds design was not good. How authors to justify this manuscript, what is the purpose to publish this manuscript?
Author Response
Reviewer 1
This manuscript described, design, synthesis, antiproliferative evaluation, docking, and MD simulation studies of 4 new VEGFR-2 inhibitors. Based on the Biological testing in the manuscript, these 4 new pounds did not show promising result, compared to sorafenib data. Based on the manuscript, I have few questions listed here.
Thank you for your efforts and your valuable comments. All comments were considered in high interest and all changes were highlighted in the revised manuscript.
- Compound 6, compound 10, C13 NMRs should be adjusted the phase line, to make the peaks to be positive position.
Response: For compounds 6 and 10 we carried out APT experiments
The phase was corrected also. In APT, CH3 and CH in one direction while CH2 and quaternary carbons in the other direction. In addition the proton data were clarified
Compound 6 C13 NMR should show 0ppm to 200ppm, H NMR should show 0ppm to 14ppm.
Response: DONE
- Compound 7 NMR in supporting information, 2.5ppm was DMSO solvent peak, 3.3ppm was the H2O residual, they should not do the integration.
Response: Done
- Compound 8 NMR in supporting information, integration should be integer.
Response: DONE
- Page 2, line 76 to line 79, authors described “sorafenib still 76 possesses some pharmacokinetic problems represented in either its bad water solubility or decreased oral bioavailability. The later problems potentiated the need for the discovery of new VEGFR-2 anti-angiogenic drugs.” But in the manuscript, the design part, did not show the logic, to solve the solubility and oral bioavailability problem. Manuscript better describe it, how authors designed the 4 compounds. Especially, the title was “Design and synthesis”.
Response: Thank you for this notice. This point was clarified as per requested.
- In vitro VEGFR-2 enzyme assay inhibition showed the new 4 compounds were not better than sorafenib. The compounds design was not good. How authors to justify this manuscript, what is the purpose to publish this manuscript?
Response: You are right that the synthesized compounds have less activity than sorafenib. But these compounds have some advantages as: i) enhanced water solubility, ii) good immunomodulatory effect, iii) good binding mode in the active pocket of VEGFR-2, iv) stability in the active pocket for 100 ns, and the simple procedure for the chemical synthesis with good yield. All these advantages encourage us to publish these compounds to be lead molecules for further modifications and optimization.
Reviewer 2 Report
Ibrahim H. Eissa, Mohamed Elzahaby and al describe the design, synthesis, anti-proliferative evaluation, docking, and MD simulation studies of new VEGFR-2 inhibitors. Since the final compounds were not fully characterized (HRMS or elemental analyses are missing), the validity of the results are doubtful. Journal titles are missinig in many references.
I encourage the resubmission of this paper
Author Response
Ibrahim H. Eissa, Mohamed Elzahaby and al describe the design, synthesis, anti-proliferative evaluation, docking, and MD simulation studies of new VEGFR-2 inhibitors. Since the final compounds were not fully characterized (HRMS or elemental analyses are missing), the validity of the results are doubtful. Journal titles are missing in many references.
Response: Elemental analyses were carried out for the synthesized compound. All obtained values were in the accepted range. Also, all references were checked and corrected in the revised manuscript.
Reviewer 3 Report
Elkaeed et al. synthesized a series of nicotinamide-based VEGFR2 inhibitors and characterized their potency, and analyzed their cell cycle, apoptotic and immunomodulatory effects in a variety of assays. The best compound in their series was compound 6, which had similar but weaker potency than sorefenib in anti-proliferative assays. Subsequently, they performed computational studies to evaluate the binding interactions for two of the most potent compounds they have synthesized. In addition, the authors evaluated the pharmacokinetic profiles of their novel inhibitors in silico and predicted them to be safe in general. These inhibitors could be potentially interesting, and their inhibitory potency could be further developed. The conclusions are supported by the presented data. However, the manuscript needs to be re-worked extensively, in particular the overall language needs to be revised and some claims need to be substantiated or clarified. Besides, there are several major issues that the authors should address before the paper can be considered for publication.
Introduction: Overall language sounds awkward. Please consider revising the language.
Line 51: Should be "especially"
Line 52: Should be “in response” not “in a response”
Line 55: Should be “produce a safe and selective”
Line 73: Typological error. Should be “hinge”, not “hing”
Line 78: Should be “latter” instead of “later”
Line 83-85: Rephrase sentence. Perhaps “of various compounds containing benzoxazole,……scaffolds”
Line 87: Should be “four new”
Line 89: The phrase should be “hinge region”.
Line 125: Should be “latter”
Line 195: Could the authors substantiate how compound 6 “is bound tightly” to VEGFR from a computational model? Can the authors comment on the degree of tight binding?
Line 201: Should it be Ile886?
Line 208: Should it be Lys866?
Line 250: Remove the brackets.
Line 285: Please elaborate on the figure legends. They are too brief.
Line 287: The word “In silico” should be in italics
Line 316: Please reduce the number of significant figures
Line 324: Please accurately phrase the table title as “Predicted toxicity parameters and properties of the synthesized compounds”, or use a similar title.
Table 1: Is there a reason why median was reported, instead of mean? Please provide more information on IC50 values. The authors should indicate whether the ± values refer to s.e.m. or s.d.? For the in vitro IC50 determination against VEGFR2, please indicate IC50 in Table 1, not “VEGFR-2 protein concentration” which is vague. Does the in vitro IC50 values have an ± values (s.e.m. or s.d.)?
Please revise the Figure legends in general and provide more detailed figure captions.
Figure 6/7/8: Amino acid residue labels were not clear. Some were obscured behind the protein cartoon structures.
Figure 7: Please make the figure legend more informative and precise.
Figure 8: Please make the figure legend more informative and precise.
Figure 9: Please label the y-axis informatively. For example, y-axis in Figure 9b) should be RMSF (nm) and Figure 9e) should be “Number of hydrogen bonds”
Figure 10: Please ensure consistent angle or positioning of the protein structures in order for readers to have a clearer idea of the changes between 1 and 100 ns
Format of numbers: Please be consistent in reporting the IC50 values or percentages. Keep to 3 significant figures if possible.
Author Response
Reviewer 3
Elkaeed et al. synthesized a series of nicotinamide-based VEGFR2 inhibitors and characterized their potency, and analyzed their cell cycle, apoptotic and immunomodulatory effects in a variety of assays. The best compound in their series was compound 6, which had similar but weaker potency than sorefenib in anti-proliferative assays. Subsequently, they performed computational studies to evaluate the binding interactions for two of the most potent compounds they have synthesized. In addition, the authors evaluated the pharmacokinetic profiles of their novel inhibitors in silico and predicted them to be safe in general. These inhibitors could be potentially interesting, and their inhibitory potency could be further developed. The conclusions are supported by the presented data. However, the manuscript needs to be re-worked extensively, in particular the overall language needs to be revised and some claims need to be substantiated or clarified. Besides, there are several major issues that the authors should address before the paper can be considered for publication.
Thank you for your praise and valuable comments. I considered these comments with high interest. The comments and my response are summarized in the following points.
- Introduction: Overall language sounds awkward. Please consider revising the language.
Response: Done
- Line 51: Should be "especially"
Response: Done
- Line 52: Should be “in response” not “in a response”
Response: Done
- Line 55: Should be “produce a safe and selective”
Response: Done
- Line 73: Typological error. Should be “hinge”, not “hing”
Response: Done
- Line 78: Should be “latter” instead of “later”
Response: Done
- Line 83-85: Rephrase sentence. Perhaps “of various compounds containing benzoxazole,……scaffolds”
Response: Done
- Line 87: Should be “four new”
Response: Done
- Line 89: The phrase should be “hinge region”.
Response: Done
- Line 125: Should be “latter”
Response: Done
- Line 195: Could the authors substantiate how compound 6 “is bound tightly” to VEGFR from a computational model? Can the authors comment on the degree of tight binding?
Response: Clarified
- Line 201: Should it be Ile886?
Response: Yes, it was corrected.
- Line 208: Should it be Lys866?
Response: it is Ile886 and it was corrected.
- Line 250: Remove the brackets.
Response: Done
- Line 285: Please elaborate on the figure legends. They are too brief.
Response: more clarifications were added into the captions of the figures
Line 287: The word “In silico” should be in italics
Response: Done
- Line 316: Please reduce the number of significant figures
Response: Done
- Line 324: Please accurately phrase the table title as “Predicted toxicity parameters and properties of the synthesized compounds”, or use a similar title.
Response: Done
- Table 1: Is there a reason why median was reported, instead of mean? Please provide more information on IC50 values. The authors should indicate whether the ± values refer to s.e.m. or s.d.? For the in vitro IC50 determination against VEGFR2, please indicate IC50 in Table 1, not “VEGFR-2 protein concentration” which is vague. Does the in vitro IC50 values have an ± values (s.e.m. or s.d.)?
Response: this typo was corrected. We used the mean and not median. In addition, we measured the IC50 of VEGFR-2 inhibition and not the protein concentration. This point was adjusted in the revised manuscript. The SD for VEGFR-2 inhibition was added.
- Please revise the Figure legends in general and provide more detailed figure captions.
Response: Done
- Figure 6/7/8: Amino acid residue labels were not clear. Some were obscured behind the protein cartoon structures.
Response: Done
- Figure 7: Please make the figure legend more informative and precise.
Response: Done
- Figure 9: Please label the y-axis informatively. For example, y-axis in Figure 9b) should be RMSF (nm) and Figure 9e) should be “Number of hydrogen bonds”
Response: Done
- Figure 10: Please ensure consistent angle or positioning of the protein structures in order for readers to have a clearer idea of the changes between 1 and 100 ns
Response: Both figures A and B were captured from the same angle.
The difference is due to the incidence of conformational changes in the VEGFR-2 enzyme
- Format of numbers: Please be consistent in reporting the IC50 values or percentages. Keep to 3 significant figures if possible.
Response: Done

Round 2
Reviewer 1 Report
My biggest concern was about the their molecular design logic, how to justify their idea.
Now, the authors added extra text and explaining. The revised manuscript is much clearer and easier to understand, now the manuscript is ready for publication. Good job.
Author Response
The authors would like to thank the reviewer for his valuable comments and questions that will make our manuscript much better.
Extensive editing for English language has been done
Reviewer 2 Report
The manuscript has been revised according to my remarks.
I thank the authors.
Author Response

(The authors gave the same response as above.)

Reviewer 3 Report
Line 50: many efforts were made to generate
Line 125: turquoise not Turquoise
Line 132: incorporation
Line 133: spacing between the and synthesized words
Line 193: Flow cytometric analysis of the cell cycle in HCT-116 after treatment with compound 6
Line 206: after treatment with 9.3 μM of compound 6 for 48 h.
Line 208: No comma
Line 236: Rephrase for clarity
Line 265: binding mode similar to that of the benzimidazole-urea inhibitor
Line 272: many HIs? Please rephrase.
Line 279: include a comma
Line 324: please correct typological error. Compounds not ompounds
Line 326: MD not M D
Line 439: confirms
Line 450: bound
Line 465: spacing needed between words are and gas. Two types
Line 517: incorrect use of the word controversy
Line 669: typological error
Line 737: the MTT procedure
Figure 10: It would be better to show superimposed structures (color differently), so that the conformational changes can be more clearly visualized. Are the conformational changes in the current figure highlighted in green?
Please ensure consistency throughout manuscript. Table X. Not Table X:
Italicize all mentions of in vitro
In the Experimental section, “In-depth discussions of….have been held in Supplementary data” is awkward sounding. Please edit.
The manuscript still contains many typological and formatting errors. I suggest the authors edit the manuscript more thoroughly.
Author Response
The authors would like to thank the Editor for his valuable help, kind guidance and professional work.
First, the authors would like to thank the reviewer for his valuable comments and questions that will make our manuscript much better.
All comments were considered and answered positively
In detail,
- Line 50: many efforts were made to generate
Response: Done
- Line 125: turquoise not Turquoise
Response: Done
- Line 132: incorporation
Response: Done
- Line 133: spacing between the and synthesized words
Response: Done
- Line 193: Flow cytometric analysis of the cell cycle in HCT-116 after treatment with compound 6
Response: Done
- Line 206: after treatment with 9.3 μM of compound 6for 48 h.
Response: Done
- Line 208: No comma
Response: Done
- Line 236: Rephrase for clarity
Response: Done
- Line 265: binding mode similar to that of the benzimidazole-urea inhibitor
Response: Done
- Line 272: many HIs? Please rephrase.
Response: Done
- Line 279: include a comma
Response: Done
- Line 324: please correct typological error. Compounds not ompounds
Response: Done
- Line 326: MD not M D
Response: Done
- Line 439: confirms
Response: Done
- Line 450: bound
Response: Done
- Line 465: spacing needed between words are and gas. Two types
Response: Done
- Line 517: incorrect use of the word controversy
Response: Done
- Line 669: typological error
Response: Done
- Line 737: the MTT procedure
Response: Done
- Figure 10: It would be better to show superimposed structures (color differently), so that the conformational changes can be more clearly visualized. Are the conformational changes in the current figure highlighted in green?
Response: Figure 10 is a Ribbon diagrams compound 6-VEGFR-2 complex at 1st and last ns. It’s a schematic representation to prove that compound 6 remained bonded correctly in the active pocked till the end of study and to illustrate the overall path and conformational changes of the compound 6-VEGFR-2 complex over the 100 ns in 3D.
We tried to superimpose both structures but the resulted figure was so confusing
- Please ensure consistency throughout manuscript. Table X. Not Table X:
Response: Done
- Italicize all mentions of in vitro
Response: Done
- In the Experimental section, “In-depth discussions of….have been held in Supplementary data” is awkward sounding. Please edit.
Response: Done
- The manuscript still contains many typological and formatting errors. I suggest the authors edit the manuscript more thoroughly.
Response: Done